# The Reactions of Alkenes with Phenyl-*N*-triflylimino-λ^3^-iodane: Solvent and Oxidant Impact

**DOI:** 10.3390/ijms242115947

**Published:** 2023-11-03

**Authors:** Mikhail Yu. Moskalik, Anton S. Ganin, Bagrat A. Shainyan

**Affiliations:** A.E. Favorsky Irkutsk Institute of Chemistry, Siberian Division of the Russian Academy of Sciences, 1 Favorsky Street, 664033 Irkutsk, Russia; moskalik@irioch.irk.ru (M.Y.M.); bagrat@irioch.irk.ru (B.A.S.)

**Keywords:** alkenes, *N*-triflylimino-λ^3^-iodane, aziridine, oxidative addition, solvent interception

## Abstract

The reactions of alkenes with phenyl-*N*-triflylimino-λ^3^-iodane PhI=NTf (**1**) have been studied in different conditions. In methylene chloride, in the presence of *N*-halosuccinimides, the products of mono and bis-triflamidation were obtained. In MeCN, the product of bromotriflamidation (with NBS) with solvent interception or of bis-triflamidation (with NIS) is formed. The reaction with *trans*-stilbene in acetonitrile with NBS gave rise to cyclization to 2-methyl-4,5-diphenyl-1-triflyl-4,5-dihydro-1*H*-imidazole. In contrast, with NIS as an oxidant, both in CH_2_Cl_2_ and MeCN, the major product was 2,3-diphenyl-1-triflylaziridine formed in good yield. With NBS, aziridine is also formed but as a minor product, the major one being a mixture of diastereomers of the product of bromotriflamidation. The reaction of compound **1** with vinylcyclohexane in methylene chloride affords the mixtures of regioisomers of the products of halotriflamidation, whereas in acetonitrile, the products of solvent interception and cyclization to the imidazoline are formed. A mechanism explaining the formation of all isolated products is proposed.

## 1. Introduction

*N*-Trifluoromethylsulfonyl (triflyl) substituted nitrogen compounds occupy a unique place in organic chemistry [1,2,3]. The triflyl group is a very strong electron-withdrawing substituent responsible for the high NH-acidity of triflamides, their specific catalytic activity, the ability to form strong intra- and intermolecular hydrogen bonds, and a number of specific chemical properties. Low nucleophilicity and high NH-acidity allow triflamides to be used in a variety of organic reactions [1,2,3].

*N*-Triflylaziridines are of considerable interest from the viewpoint of their further transformations, allowing us to obtain structures that already contain the triflyl group. They have wide areas of application, such as nucleophilic ring opening to give *N*-protected alkyleneamines [4] and catalytic enantioselective alkylation of the methine carbon of carboxylic acids [5], which are used in the synthesis of tripodal tetradentate C3-symmetric amines [6]. *N*-Triflylaziridines are also widely employed in the synthesis of ligands for metal complex catalysis [7,8,9,10,11,12,13,14]. However, the methods of their synthesis meet significant difficulties precisely because of specific properties of the triflyl group.

Prior to our studies, several examples of the synthesis of *N*-triflylaziridines were described in the literature. One of the first examples studied was the classical reaction of triflylazide with electron-rich *O*-TMS vinyl ethers [15] (Figure 1, route 1):

The necessity to use an unstable and explosive triflyl azide greatly complicates the reaction. Similar transformations of perfluorosulfonylazides and O-TMS ethers via the formation of *N*-perfluoromethylsulfonylaziridines give a set of α-[*N*-(peruoroalkanesulfonyl)]aminoketones and α-amino acids in good yields [16,17]. UV activation of the reaction of butene-2 with the azide H(CF_2_)_2_O(CF_2_)_2_SO_2_N_3_ affords the corresponding aziridine in 61% yield [18].

*N*-Triflylaziridine was also obtained by treating (2*S*)-2-amino-3-methylbutan-1-ol with triflic anhydride on strong cooling (Figure 1, route 2) [19].

One of the most efficient methods for the synthesis of *N*-triflylaziridines is the reaction of alkenes with *N*-triflylimino-λ^3^-bromane 4-CF_3_C_6_H_4_Br=NTf [20,21] prepared in situ or in advance through several steps [21,22] (Figure 1, route 3). The reagent is formed by the reaction of triflamide with 4-CF_3_C_6_H_4_Br(OAc)_2_ synthesized, in turn, by treating difluoride 4-CF_3_C_6_H_4_BrF_2_ with acetic anhydride [21]. Difluoride is obtained by the reaction of 4-CF_3_C_6_H_4_SiMe_3_ with BrF_3_ [22]. *N*-Triflylimino-λ^3^-bromane is relatively stable and can be stored under argon at 4 °C for several months [21,22]. The reagent can also be obtained directly, bypassing the diacetate formation step, from 4-CF_3_C_6_H_4_BrF_2_ [23]. Aziridination occurs with high chemoselectivity and in good to excellent yields (75–98%) (Figure 1, Pathway 2) [20,21]. The scope of substrates included cycloalkenes C_6_–C_8_, norbornene, styrenes, and octenes. At the same time, the overall synthesis of *N*-triflylaziridines in Figure 1, Pathway 2 involves four separate stages and the use of BrF_3_, which significantly limits the synthetic applicability of the method. 

The addition of *N*,*N*-dichloroalkanesulfonamide I(CF_2_)_2_O(CF_2_)_2_SO_2_NCl_2_ to styrene in the presence of EtONa/EtOH affords the corresponding aziridine in 49% yield [24]. A shortcoming of highly reactive *N*,*N*-dichloroalkanesulfonamides as reagents is their instability in air moisture and the need for special storage conditions [25].

More stable than triflylimino-λ^3^-bromane are triflylimino-λ^3^-iodanes. The *N*-triflyl-substituted analogs of the latter, *N*-triflylimino-λ^3^-iodanes ArI=NTf, were synthesized by the reaction of iodosobenzenes ArI=O with triflamide as late as in 2022 and used for amidation or imidation of phosphines and 1,3-diketones [26]. However, in our recent work [27], we found no aziridines in the reaction of styrene with triflamide and iodosobenzene in the presence of iodine and CuCl. 

## 2. Results and Discussion

Therefore, the synthesis of *N*-triflylaziridines remains a challenging task. Based on this, we have studied the reactions of alkenes with phenyl-*N*-triflylimino-λ^3^-iodane PhI=NTf (**1**) in different conditions. In methylene chloride, in the absence of oxidants, no reactions occurred with styrene, vinylcyclohexane, dimethyl(divinyl)silane, or tetramethyl(divinyl)siloxane; only the starting reagent **1** was recovered. Given that alkenes readily undergo triflamidation with solvent interception affording *N*-triflylamidines in the reaction with triflamide in the presence of *N*-halosuccinimides [27,28,29], we have studied the reactions of compound **1** with styrene in the presence of *N*-bromo (NBS) and *N*-iodosuccinimide (NIS). Depending on the solvent, the products of mono and bis-triflamidation of the double bond were obtained. Note that in the products of halotriflamidation **3**, **4** the triflamide residue adds to the α-carbon of the styrene molecule (Figure 2). 

The structure of the products and, hence, the regioselectivity of addition is proved by the presence of the NH signal at 5.76 (**3**) and 5.89 ppm (**4**), multiplets of the methine proton at 4.97 (**3**) and 4.78 ppm (**4**), and doublets of doublets at 3.74 (**3**) and 3.56 (**4**) of the methylene protons.

Varying the solvent from CH_2_Cl_2_ to MeCN changes the course of the reaction; with NBS, *N*-(2-bromo-1-phenylethyl)-*N*′-(triflyl)acetimidamide **6** is obtained in moderate yield (Figure 3).

The structure of amidine **6** was confirmed by the presence of the NH doublet at 7.67 ppm, a doublet of triplets of CHN at 5.34 ppm, two doublets of diastereotopic CH_2_Br at 3.73 and 3.64 ppm, and the Me group singlet at 2.48 ppm. The ^13^C NMR spectrum contains the signals at 169 (C=N), 58 (CHN), 34 (CH_2_Br), and 22.3 (Me). The NMR spectra for compounds **3**, **4**, **6** is offered in Appendix A.

In the presence of NIS, styrene reacts with *N*-triflylimino-λ^3^-iodane **1** in acetonitrile to give the product of bis-triflamidation TfNHCH(Ph)–CH_2_NHTf **7** in 54% yield, identical to that obtained earlier by the reaction with triflamide in the system *t*-BuOCl/NaI [30]. 

In continuation, we examined the reaction of compound **1** with some alkenes in the system I_2_/NaI/MeCN, which was successfully employed for the synthesis of *N*-tosylaziridines [31] (*vide supra*). In contrast, no aziridine was formed with compound **1**, but only the aforementioned adduct **7** in a low yield of 25%. Under the same conditions, *trans*-stilbene and vinylcyclohexane did not react with compound **1**. UV activation did not change the course of the reaction and only increased the yield of adduct **7** to 40%.

When the oxidant I_2_/NaI was replaced by NBS or NIS, the reaction of *N*-triflylimino-λ^3^-iodane **1** and *trans*-stilbene **8** in acetonitrile led to cyclization and formation of 2-methyl-4,5-diphenyl-1-triflyl-4,5-dihydro-1*H*-imidazole **9** (Figure 4).

The structure and composition of imidazole **9** was confirmed by IR, NMR spectroscopy, and high-resolution mass spectrometry (HRMS), which showed the molecular ion [M + H]^+^ with *m/z* 369.08807 corresponding to C_17_H_16_F_3_N_2_O_2_S^+^. When using NIS as the oxidant, imidazole **9** was also formed but as a minor product, the major one being 2,3-diphenyl-1-triflylaziridine **10** isolated in 78% yield (Figure 5). The formation of aziridine **10** is the first case of the synthesis of *N*-sulfonylaziridines using *N*-sulfonylimino-λ^3^-iodanes as a source of the nitrogen group. 

The structure of aziridine **10** was proved by IR and NMR spectroscopy and confirmed by HRMS data. In particular, the values of the ^1^*J*_CH_ constants in the aziridine moiety are known to be much larger than in saturated aliphatic compounds and lie in the range of 170–180 Hz [32,33,34]. The measured value of ^1^*J*_CH_ in product **10** is 177 Hz, which unambiguously proves its structure. The HRMS spectrum shows a molecular ion *m/z* at 328.06242, corresponding to the molecular formula C_15_H_13_F_3_NO_2_S^+^.

Earlier, it was shown that the reaction of styrenes with tosylamide in acetonitrile in the presence of *t*-BuOI leads to the substituted aziridines [35]. However, no aziridine is formed in the reaction with triflamide under the same conditions, but only the product of bis-triflamidation and disubstituted piperazine, which is, formally, the product of dimerization of the target aziridine [30]. It can be assumed that the formation of piperazine is the result of relative instability of the intermediate 2-phenyl-1-triflylaziridine, because its analog, 2-phenyl-1-tosylaziridine is stable [30]. Presumably, the stability of aziridine **10** is due to the presence of the second phenyl group stabilizing the molecule. In methylene chloride, the NIS-induced reaction also gives aziridine **10** in about the same yield, but with NBS it is the minor product, whereas the major product was identified to be *N*-(2-bromo-1,2-diphenylethyl)triflamide **11** (Figure 6).

According to the NMR spectroscopy data, compound **11**, possessing two chiral carbon atoms, is formed as a mixture of two pairs of diastereomers. Two sets of the NH, CHBr, and CHN signals and two pairs of the CHBr and CHN ^13^C signals appear in the ratio of 5:1. The NMR and HRMS spectra for compounds **9**–**11** is offered in Appendix A.

The reaction of vinylcyclohexane **12** with *N*-triflylimino-λ^3^-iodane **1** in methylene chloride in the presence of NBS or NIS results in a mixture of the regioisomers of the products of bromo- or iodotriflamidation in the ratio of 1:2 (**13**:**14** = 21:41%) or 1:3 (**15**:**16** = 14:43%) and total yield of 62 or 57% (Figure 7):

The structure of the regioisomers was assigned based on the chemical shifts and multiplicity of the signals in the ^1^H and ^13^C NMR spectra. The NH signals in compounds **13** and **15** appear as clearly resolved doublets, whereas in their isomers **14** and **16**, they look as an unresolved broad singlet (X = Br) or as a triplet (X = I). In regioisomers **13** and **15** the methine protons of the CH–CH_2_ spin system resonate in a higher field than the methylene protons, whereas in regioisomers **14** and **16** their relative position is reversed. In the ^13^C NMR spectra, the most upfield signals belong to the CH_2_Br (37 ppm) or CH_2_I carbons (13 ppm).

Replacing the solvent in the reaction in Figure 7 from methylene chloride to acetonitrile gives rise to the formation of additional products. In the presence of NIS, apart from regioisomers **15** and **16**, the isomeric amidines **17** and **18** were isolated as the products of solvent interception (Figure 8):

In a similar way the reaction proceeds in the presence of NBS, except for the formation of imidazoline **19** isomeric to the earlier synthesized 5-cyclohexyl-2-methyl-1-triflyl-4,5-dihydro-1*H*-imidazole [27] and the product of solvent interception/hydrolysis **20** in the total yield of 88% (Figure 9):

The NMR and HRMS spectra for compounds **13**–**19** is offered in Appendix A.

A mechanistic scheme allowing to explain the formation of all products in Figure 2, Figure 3, Figure 4, Figure 5, Figure 6, Figure 7, Figure 8 and Figure 9 is presented in Figure 10. The specific reactivity of triflamide leads to the formation (under the conditions of the reactions) of both *N*-sulfonylamidines, products containing acetamide and triflamide groups, and products formed without the participation of a solvent as a reagent in the reaction. Since the basicity of acetonitrile (780 kJ/mol) [36] is higher than that of triflamide (740 kJ/mol) [1], acetonitrile is a stronger nucleophile than triflamide, which leads to the formation of amidines (or the corresponding acetamides upon hydrolysis of the corresponding intermediate), as shown in Figure 10. The use of CH_2_Cl_2_ eliminates the involvement of the solvent in the reaction.

Figure 10 is consistent with the formation of *N*-triflylaziridine **10** (instead of the corresponding amidine) for the best nucleofuge (X = I) even in MeCN, as well as with a much higher yield of **10** for X = I than for X = Br when performing the reaction in CH_2_Cl_2_ (Figure 10): 

## 3. Materials and Methods

### 3.1. General Details 

The ^1^H (400.1 MHz), ^13^C (100.6 MHz), and ^19^F (376.0 MHz) NMR spectra were registered on a Bruker DPX-400 spectrometer for 5–10% solutions in CDCl_3_. ^1^H and ^13^C chemical shifts (δ) are reported in parts per million (ppm) relative to the residual solvent peak of CDCl_3_ (δ = 7.27 (^1^H) and 77.10 (^13^C) ppm, respectively). ^19^F NMR chemical shifts (δ) are given relative to CCl_3_F. All coupling constants (*J*) are reported in hertz (Hz). Abbreviations are s, singlet; d, doublet; t, triplet; q, quartet; and brs, broad singlet. The IR spectra (cm^−1^) were taken on a Bruker Vertex 70 spectrometer. Mass spectra were registered in ESI-TOF-HRMS mode on an Agilent 6210 instrument (Agilent Technologies, Santa Clara, CA, USA). Elemental composition was determined on a Thermo Scientific Flash 2000 CHNS analyzer. Melting points were measured on a Boetius apparatus. TLC was performed on silica 60 plates (0.25 mm, F254, Merck, Rahway, NJ, USA) and visualized by UV lamp. Commercial reagents and solvents were used without further purification unless otherwise mentioned.

### 3.2. Synthesis of Compounds 

#### 3.2.1. Synthesis of Phenyl-*N*-triflylimino-λ^3^-iodane **1**

Triflamide (0.50 g, 3.4 mmol) was added to a solution of iodosolbenzene (0.75 g, 3.4 mmol) in methylene chloride (10 mL) and stirred for 5 min. Then, the solvent was distilled off to dryness in a vacuum, and the solid white residue was washed with hexane to remove triflamide. The residue was dried in a vacuum to obtain 1.10 g (94%) iminoiodane **1**.

#### 3.2.2. Reaction of PhI=NTf **1** with Styrene in the System NXS–CH_2_Cl_2_

To 15 mL of freshly distilled CH_2_Cl_2_, styrene (0.208 g, 2 mmol) and compound **1** (0.351 g, 1 mmol) were added. The mixture was stirred for 5 min, then NBS (0.178 g, 1 mmol) or NIS (0.225 g, 1 mmol) was added, stirred for 24 h, and the solvent was removed. The residue was washed from unreacted reagents with hexane–ethyl acetate (4:1), the solvent was removed, and the oily residue was purified by column chromatography (silica, 0.06–0.20 mm, Acros Organics) eluting with ether–hexane 1:2 mixture to get *N*-(2-halo-1-phenylethyl)triflamides **3**, **4**, and 1:1 mixture to obtain 2-iodo-1-phenylethan-1-ol **5**.

*N*-(2-Bromo-1-phenylethyl)triflamide (**3**) 

Yellow oil, 0.209 g (63%). IR ν_max_ (thin, cm^−1^): 3303 (NH), 3092, 3067, 3035, 2919, 2852, 2373, 1721, 1602, 1548, 1496, 1455, 1429, 1377, 1231, 1196, 1145, 1067, 1017, 961, 845, 816, 788, 759, 701, 615, 565, 511. ^1^H NMR (CDCl_3_): δ 7.40 (m, 3H, Ph), 7.30 (d, *J* = 7.4 Hz, 2H, Ph), 5.76 (br.s, 1H, NH), 4.97 (q, *J* = 5.9 Hz, 1H, CHN), 3.78 (dd, *J* = 10.7, 5.3 Hz, 1H, CH_2_Br), 3.70 (dd, *J* = 10.9, 5.8 Hz, 1H, CH_2_Br). ^13^C NMR (CDCl_3_): δ 137.04 (C*^u^*), 129.22 (C*^m^*), 129.13 (C*^p^*), 126.26 (C*^o^*), 121.25 (q, *J* = 321.2 Hz, CF_3_), 58.99 (CHN), 36.49 (CH_2_Br). ^19^F NMR (CDCl_3_): δ_F_ = −77.02. Anal., found: C, 32.60; H, 2.75; N, 4.26; F, 17.20; S, 9.70. Calcd for C_9_H_9_BrF_3_NO_2_S: C, 32.55; H, 2.73; N, 4.22; F, 17.16; S, 9.65.

*N*-(2-Iodo-1-phenylethyl)triflamide (**4**) 

Yellow-orange oil, 0.195 g (51%). IR ν_max_ (thin, cm^−1^): 3297 (NH), 3091, 3066, 3033, 2958, 2920, 1722, 1708, 1693, 1661, 1646, 1628, 1602, 1588, 1547, 1526, 1496, 1455, 1424, 1376, 1270, 1231, 1198, 1143, 1064, 1030, 1003, 945, 915, 878, 841, 761, 699, 652, 609, 588, 508. ^1^H NMR (CDCl_3_): δ 7.42 (m, 3H, Ph), 7.29 (d, *J* = 7.7 Hz, 2H, Ph), 5.89 (d, *J* = 8.3 Hz, 1H, NH), 4.78 (dt, *J* = 8.0, 6.5 Hz, 1H, CHN), 3.60 (dd, *J* = 10.7, 6.1 Hz, 1H, CH_2_I), 3.53 (dd, *J* = 10.2, 5.9 Hz, 1H, CH_2_I). ^13^C NMR (CDCl_3_): δ 137.82 (C*^u^*), 129.19 (C*^m^*), 129.05 (C*^p^*), 126.04 (C*^o^*), 121.20 (q, *J* = 321.1 Hz, CF_3_), 59.33 (CHN), 10.40 (CH_2_I). ^19^F NMR (CDCl_3_): δ_F_ −76.95. Anal., found: C, 28.63; H, 2.50; N, 3.73; F, 15.10; S, 8.53. Calcd for C_9_H_9_IF_3_NO_2_S: C, 28.51; H, 2.39; N, 3.69; F, 15.03; S, 8.46.

2-Iodo-1-phenylethan-1-ol (**5**) 

Violet liquid, 0.06 g (12%). Identical to that in [30].

#### 3.2.3. Reaction of PhI=NTf **1** with Styrene in the System NXS–MeCN

The reaction was performed and treated as above. To the residue, ether was added, kept for 1 h in a fridge, and the precipitated succinimide was filtered off. The ether was removed, and the oily residue was purified by column chromatography eluting with a 1:2 or 2:1 mixture to obtain products **6** and **7**.

*N*-(2-Bromo-1-phenylethyl)-*N*′-(triflyl)acetimidamide (**6**) 

Yellow oil, 0.263 g (70%). IR ν_max_ (thin, cm^−1^): 3216 (NH), 3065, 3034, 2954, 2924, 2854, 2649, 2595, 1958, 1885, 1804, 1740, 1654, 1590, 1554, 1495, 1486, 1452, 1432, 1377, 1325, 1229, 1197, 1148, 1093, 1055, 1006, 987, 920, 863, 832, 759, 700, 667, 607, 582, 538, 507. ^1^H NMR (CDCl_3_): δ 7.67 (d, J = 7.0 Hz, 1H, NH), 7.35 (m, 3H, Ph), 7.28 (dd, J = 7.4, 2.2 Hz, 2H, Ph), 5.34 (dt, J = 7.7, 5.4 Hz, 1H, CHN), 3.73 (dd, J = 10.7, 8.3 Hz, 1H, CH_2_Br), 3.64 (dd, J = 11.0, 5.1 Hz, 1H, CH_2_Br), 2.48 (s, 3H, CH_3_). ^13^C NMR (CDCl_3_): δ 168.71 (C=N), 137.05 (C*^u^*), 129.41 (C*^m^*), 128.94 (C*^p^*), 126.82 (C*^o^*), 121.17 (q, *J* = 320.9 Hz, CF_3_), 57.67 (CHN), 33.56 (CH_2_Br), 21.88 (CH_3_). ^19^F NMR (CDCl_3_): δ_F_ −79.14. Anal., found: C, 35.46; H, 3.30; N, 7.60; F, 15.30; S, 8.70. Calcd for C_10_H_15_BrF_3_NO_3_S: C, 35.40; H, 3.24; N, 7.51; F, 15.27; S, 8.59.

*N*,*N*′-(1-Phenylethane-1,2-diyl)bis(triflamide) (**7**) 

White powder, 0.215 g (54%). Identical to that described earlier [30].

#### 3.2.4. Reaction of PhI=NTf **1** with Styrene in the System I_2_–NaI–MeCN

To 15 mL of freshly distilled MeCN, styrene (0.208 g, 2 mmol) and PhI=NTf (0.351 g, 1 mmol) were added, stirred for 5 min, and added I_2_ (25 mg, 0.1 mmol) and NaI (8 mg, 0.05 mmol). The mixture was stirred for 24 h and poured into aqueous solution of Na_2_S_2_O_3_. The obtained solution was extracted with ether, dried over anhydrous MgSO_4_, and ether removed in a vacuum. The oily residue was purified by column chromatography eluting with ether-hexane 1:2 to obtain product **7** (0.126 g, 32%). 

#### 3.2.5. Reaction of PhI=NTf **1** with Stilbene in the System NXS-MeCN

To 10 mL of freshly distilled MeCN, stilbene (0.180 g, 1 mmol) and PhI=NTf (0.351 g, 1 mmol) were added, stirred for 5 min, and NBS (0.178 g, 1 mmol) or NIS (0.225 g, 1 mmol) was added. The mixture was stirred for 24 h, the solvent removed, the residue was mixed with ether, kept for 1 h in a fridge, and the precipitated succinimide was filtered off. The ether was evaporated, and the oily residue was purified by column chromatography, eluting with hexane to obtain 2,3-diphenyl-1-(triflyl)aziridine **10** and with the mixture ether-hexane 1:2 to obtain 2-methyl-4,5-diphenyl-1-triflyl-4,5-dihydro-1*H*-imidazole **9**.

2-Methyl-4,5-diphenyl-1-((trifluoromethyl)sulfonyl)-4,5-dihydro-1*H*-imidazole (**9**)

Colorless oil, 0.264 g (72%). IR ν_max_ (thin, cm^−1^): 3373, 3308, 3170, 3089, 3066, 3034, 2919, 2852, 1957, 1664, 1603, 1587, 1549, 1496, 1455, 1434, 1405, 1386, 1364, 1227, 1213, 1198, 1153, 1096, 1079, 1062, 1015, 981, 951, 915, 786, 759, 698, 645, 607, 582, 534. ^1^H NMR (CDCl_3_): δ 7.41 (m, 6H, Ph), 7.9 (d, *J* = 7.0 Hz, Ph), 7.17 (d, *J* = 7.0 Hz, 2H, Ph), 5.12 (d, *J* = 2.3 Hz, 1H, CHN), 5.06 (d, *J* = 2.9 Hz, 1H, CHN), 2.56 (s, 3H, CH_3_). ^13^C NMR (CDCl_3_): δ 154.73 (C=N), 140.36 (C*′^i^*), 139.98 (C*^i^*), 129.36 (C′*^m^*), 129.21 (C′*^o^*), 129.03 (C′*^p^*) 128.48 (C*^p^*), 126.23 (C*^m^*), 125.91 (C*^o^*), 121.40 (q, CF_3_, *J* = 323.3 Hz), 77.63 (CHN), 72.75 (CHN), 16.61 (CH_3_). ^19^F NMR (CDCl_3_): δ_F_ −79.05. HRMS (ESI): *m/z*: [M + H]^+^ calcd for C_17_H_16_F_3_N_2_O_2_S^+^: 369.08845; found: 369.08807.

2,3-Diphenyl-1-(triflyl)aziridine (**10**)

Colorless oil, 0.254 g (78%). IR ν_max_ (thin, cm^−1^): 3305, 3087, 3063, 3033, 2983, 2922, 2852, 2362, 1957, 1888, 1810, 1726, 1657, 1600, 1579, 1540, 1495, 1453, 1406, 1378, 1317, 1279, 1226, 1201, 1148, 1071, 1028, 1002, 943, 916, 871, 797, 749, 698, 638, 613, 513. ^1^H NMR (CDCl_3_): δ 7.37 (m, 5H, Ph), 3.88 (s, 1H, CH). ^13^C NMR (CDCl_3_): δ 137.22 (C*^i^*), 128.65 (C*^m^*), 128.40 (C*^p^*), 125.60 (C*^o^*), 121.36 (q, CF_3_, *J* = 307.4 Hz), 62.91 (d, CHN, *J* = 177.7 Hz). ^19^F NMR (CDCl_3_): δ_F_ = −75.99. Anal., found: C, 35.46; H, 3.30; N, 7.60; F, 15.30; S, 8.70. Calcd for C_10_H_15_BrF_3_NO_3_S, %: C, 35.40; H, 3.24; N, 7.51; F, 15.27; S, 8.59. HRMS (ESI): *m/z*: [M + H]^+^ calcd for C_15_H_13_F_3_NO_2_S^+^: 328.06191; found: 328.06242.

#### 3.2.6. Reaction of PhI=NTf **1** with Stilbene in the System NBS(NIS)-CH_2_Cl_2_

The reaction was performed and treated as above. After the removal of the solvent, the solid residue was washed with the mixture hexane–ethyl acetate (4:1) to separate the unreacted compounds. The liquid residue was kept in a vacuum, the oily residue was purified by column chromatography eluting with hexane to obtain 2,3-diphenyl-1-triflylaziridine **10**, and with a mixture of ether-hexane (1:2) to isolate *N*-(2-bromo-1,2-diphenylethyl)triflamide **11**.

*N*-(2-Bromo-1,2-diphenylethyl)triflamide (**11**)

Colorless oil, 0.220 g (45%). IR ν_max_ (thin, cm^−1^): 3282 (NH), 3108, 3090, 3066, 3011, 2982, 2922, 2854, 1961, 1885, 1807, 1716, 1601, 1588, 1496, 1455, 1367, 1277, 1228, 1197, 1076, 1030, 1002, 972, 940, 916, 866, 818, 762, 698, 613. ^1^H NMR (CDCl_3_): δ 7.31 (m, 5H, Ph), 7.09 (m, 2H, Ph), 7.01 (m, 2H, Ph), 5.66 (d, *J* = 9.5 Hz, 1H, NH) 5.35 (d, *J* = 5.0 Hz, 1H, CHBr), 5.03 (dd, *J* = 9.5, 5.0 Hz, 1H, CHN). ^13^C NMR (CDCl_3_): δ 135.78 (C*^i^*), 135.77 (C*^i^*^′^), 129.29 (C*^p^*), 129.00 (C*^p^*^′^), 128.84 (C*^m^*), 128.63 (C*^o^*), 128.41 (C*^m^*^′^), 127.59 (C*^o^*^′^), 119.48 (q, CF_3_, *J* = 321.4 Hz), 64.30 (CHBr), 58.08 (CHN). ^19^F NMR (CDCl_3_): δ_F_ = −77.10. HRMS (ESI): *m/z*: [*M*–Br]^+^ calcd for C_15_H_13_F_3_NO_2_S^+^: 328.061908; found: 328.062180.

#### 3.2.7. Reaction of PhI=NTf **1** with Vinylcyclohexane in the System NBS(NIS)-CH_2_Cl_2_

The reaction was performed and treated as above. The solid residue was washed with hexane–ethyl acetate (4:1) to separate unreacted reagents. The solvent was evaporated, and the oily residue was purified by column chromatography eluting with ether-hexane (1:2) to obtain the mixture of regioisomers **13** and **14**, or **15** and **16**.

*N*-(2-Bromo-1-cyclohexylethyl)triflamide (**13**) 

Colorless oil, 0.070 g (21%). IR ν_max_ (thin, cm^−1^): 3321, 3062, 2926, 2855, 1656, 1516, 1444, 1430, 1379, 1225, 1196, 1148, 1080, 1051, 1024, 993, 956, 914, 893, 854,791, 761, 644, 604, 564. ^1^H NMR (CDCl_3_): δ 5.15 (d, *J* = 8.5 Hz, 1H, NH), 3.65 (d, *J* = 2.8 Hz, 1H, CH_2_Br), 3.61 (d, *J* = 3.7 Hz, 1H, CH_2_Br), 3.43 (m, 1H, CHN), 1.74–1.57 (m, 5H), 1.29–1.11 (m, 6H). ^13^C NMR (CDCl_3_): δ 119.69 (q, CF_3_, *J* = 320.5 Hz), 60.12 (CHN), 39.54 (CH), 36.07 (CH_2_Br), 29.23, 28.90, 25.87, 25.74, 25.71 (CH_2_). ^19^F NMR (CDCl_3_): δ_F_ −77.23.

*N*-(2-Bromo-2-cyclohexylethyl)triflamide (**14**) 

Colorless oil, 0.140 g (41%). IR ν_max_ (thin, cm^−1^): 3326, 3069, 2926, 2855, 1706, 1656, 1516, 1448, 1431, 1374, 1229, 1195, 1149, 1080, 1050, 1025, 993,958, 917, 892, 851,791, 761, 644, 605, 562. ^1^H NMR (CDCl_3_): δ 5.46 (br.s, 1H, NH), 3.99 (dt, *J* = 8.9, 4.0 Hz, 1H, CHBr), 3.73 (dd, *J* = 14.4, 3.1 Hz, 1H, CH_2_N), 3.54 (dd, *J* = 14.4, 9.2 Hz, 1H, CH_2_N), 1.85–1.72 (m, 5H), 1.29–1.14 (m, 6H). ^13^C NMR (CDCl_3_): δ 121.62 (q, CF_3_, *J* = 318.8 Hz), 62.34 (CHBr), 48.72 (CH_2_N), 42.05 (CH), 30.67, 29.85, 26.01, 25.90, 25.80 (CH_2_). ^19^F NMR (CDCl_3_): δ_F_ −77.11. Anal., found: C, 31.98; H, 4.50; N, 4.17; F, 16.90; S 9.51. Calcd for C_9_H_15_BrF_3_NO_2_S: C, 31.96; H, 4.47; N, 4.14; F, 16.85; S, 9.48.

*N*-(1-Cyclohexyl-2-iodoethyl)triflamide (**15**) 

Light-brown oil, 0.055 g (14%). IR ν_max_ (thin, cm^−1^): 3303, 2928, 2855, 2798, 2341, 1719, 1429, 1378, 1233, 1192, 1144, 1098, 1071, 1025, 991, 952, 920, 890, 849, 783, 632, 607, 577. ^1^H NMR (CDCl_3_): δ 4.91 (d, *J* = 8.9 Hz, 1H, NH), 3.45 (d, *J* = 3.4 Hz, 2H, CH_2_I), 2.92 (dq, *J* = 8.3, 3.4 Hz, 1H, CHN), 1.72–1.63 (m, 5H), 1.33–1.14 (m, 6H). ^13^C NMR (CDCl_3_): δ 119.63 (q, CF_3_, *J* = 320.9 Hz), 59.34 (CHN), 41.40 (CH), 29.10, 28.55, 25.85, 25.68 (CH_2_), 12.07 (CH_2_I). ^19^F NMR (CDCl_3_): δ_F_ −77.18.

*N*-(2-Cyclohexyl-2-iodoethyl)triflamide (**16**) 

Light-brown oil, 0.165 g (43%). IR ν_max_ (thin, cm^−1^): 3308, 2931, 2857, 2364, 1733, 1448, 1372, 1233, 1193, 1144, 1096, 1080, 1041, 993, 954, 920, 890, 851, 783, 632, 607, 577. ^1^H NMR (CDCl_3_): δ 5.30 (t, *J* = 6.3 Hz, 1H, NH), 4.10 (dt, *J* = 8.3, 4.8 Hz, 1H, CHI), 3.63 (t, *J* = 4.9 Hz, 2H, CH_2_N), 1.83–1.74 (m, 5H), 1.34–1.08 (m, 6H). ^13^C NMR (CDCl_3_): δ 119.79 (q, CF_3_, *J* = 321.1 Hz), 50.01 (CHI), 44.94 (CH_2_N), 42.01 (CH), 32.42, 31.67, 26.05, 25.76 (CH_2_). ^19^F NMR (CDCl_3_): δ_F_ −77.02. Anal., found: C, 28.10; H, 3.95; N, 3.67; F, 14.82; S 8.35. Calcd for C_9_H_15_IF_3_NO_2_S: C, 28.06; H, 3.93; N, 3.64; F, 14.80; S, 8.32.

#### 3.2.8. Reaction of PhI=NTf **1** with Vinylcyclohexane in the System NIS-MeCN

The reaction was performed and treated as above. The residue after separation of succinimide and evaporation of ether was purified by column eluting with hexane to obtain a mixture of regioisomers **15** and **16** (50 mg), then with ether-hexane (4:1), to obtain the regioisomers of amidines **17** and **18**. 

*N*-(1-Cyclohexyl-2-iodoethyl)-*N*′-(triflyl)acetimidamide (**17**) 

Brown liquid, 0.133 g (31%). IR ν_max_ (thin, cm^−1^): 3299, 3065, 2932, 2857, 2672, 1655, 1550, 1440, 1420, 1380, 1301, 1280, 1232, 1198, 1145, 1072, 1050, 1022, 991, 964, 957, 891, 852, 787, 645, 608. ^1^H NMR (CDCl_3_): δ 6.64 (d, *J* = 8.8 Hz, 1H, NH), 4.09 (dt, *J* = 7.6, 3.4 Hz, 1H, CHN), 3.44 (dd, *J* = 10.8, 3.6 Hz, 1H, CH_2_I), 3.30 (dd, *J* = 10.9, 5.4 Hz, 1H, CH_2_I), 2.51 (s, 3H, CH_3_), 1.89–1.61 (m, 6H), 1.35–1.07 (m, 5H). ^13^C NMR (CDCl_3_): δ 169.02 (C=N), 119.77 (q, CF_3_, *J* = 319.2 Hz), 56.76 (CHN), 41.56 (CH), 32.21, 31.70, 25.68, 25.65 (CH_2_), 22.15 (CH_3_), 8.85 (CH_2_I). ^19^F NMR (CDCl_3_): δ_F_ −78.86. Anal., found: C, 31.04; H, 4.29; N, 6.59; F, 13.39; S, 7.53. Calcd. for C_11_H_18_F_3_IN_2_O_2_S: C, 31.00; H, 4.26; N, 6.57; F, 13.37; S, 7.52.

*N*-(2-Cyclohexyl-2-iodoethyl)-*N*′-((trifluoromethyl)sulfonyl)acetimidamide (**18**)

Brown liquid, 0.10 g (24%). IR ν_max_ (thin, cm^−1^): 3306, 3069, 2932, 2857, 2672, 1732, 1655, 1518, 1449, 1430, 1377, 1304, 1279, 1232, 1196, 1147, 1079, 1050, 1030, 994, 962, 919, 893, 853, 789, 645, 608. ^1^H NMR (CDCl_3_): δ 6.47 (br.s, 1H, NH), 4.20 (dt, *J* = 9.7, 4.0 Hz, 1H, CHI), 3.94 (ddd, *J* = 14.5, 6.4, 3.6 Hz, 1H, CH_2_N), 3.55 (ddd, *J* = 14.5, 9.7, 4.9 Hz, 1H, CH_2_N), 2.52 (s, 3H, CH_3_), 1.85–1.62 (m, 6H), 1.35–1.09 (m, 5H). ^13^C NMR (CDCl_3_): δ 168.59 (C=N), 119.88 (q, CF_3_, *J* = 320.6 Hz), 48.65 (CH_2_N), 43.82 (CH), 43.17 (CHI), 32.05, 32.03, 26.09, 25.91 (CH_2_), 22.47 (CH_3_). ^19^F NMR (CDCl_3_): δ_F_ −79.07. HRMS (ESI): *m/z*: [*M*–I]^+^ calcd for C_11_H_18_F_3_N_2_O_2_S ^+^: 299.104109; found: 299.104370.

#### 3.2.9. Reaction of PhI=NTf **1** with Vinylcyclohexane in the System NBS-MeCN

The reaction was performed and treated as above. The residue after solvent removal was poured with ether, kept for 1 h in a fridge, and the precipitated succinimide was filtered off. The ether was evaporated, and the oily residue was purified by column eluting with hexane to afford **14** (0.05 g), then with ether-hexane (1:2) to obtain **20**, then with ether-hexane (2:1) to obtain **19** and, finally, with ether-hexane (4:1) to get **21**.

4-Cyclohexyl-2-methyl-1-triflyl-4,5-dihydro-1*H*-imidazole (**19**) 

White powder, 0.105 g (35%). Mp = 163.6 °C. IR ν_max_ (thin, cm^−1^): 3013, 2933, 2637, 2588, 1675, 1638, 1483, 1449, 1415, 1378, 1271, 1235, 1207, 1156, 1092, 1047, 1030, 979, 896, 864, 843, 772, 685, 619, 580, 531. ^1^H NMR (CDCl_3_): δ 4.51 (dd, *J* = 15.2, 10.7 Hz, 2H, CH_2_N), 4.08 (dd, *J* = 15.6, 12.8 Hz, 1H, CHN), 2.89 (s, 3H, CH_3_), 1.88–1.59 (m, 7H), 1.34–1.11 (m, 4H). ^13^C NMR (CDCl_3_): δ 168.95 (C=N), 118.59 (q, *J* = 323.6 Hz, CF_3_), 63.14 (CHN), 52.81 (CH_2_N), 41.03 (CH), 28.36, 27.93, 25.64, 25.76, 25.30 (CH_2_), 14.73 (CH_3_). ^19^F NMR (CDCl_3_): δ_F_ −74.13. HRMS (ESI): *m/z*: [M + H]^+^ calcd for C_11_H_18_F_3_N_2_O_2_S^+^: 299.104109; found: 299.10376.

*N*-(1-cyclohexyl-2-(triflamidoethyl)acetamide (**20**) (white powder, 0.03 g (12%) and *N*-(2-bromo-2-cyclohexylethyl)-*N*′-(triflyl)acetimidamide (**21**) (yellow liquid, 0.092 g (24%) are identical to the obtained and characterized earlier [27].

## 4. Conclusions

To summarize, the reactions of alkenes with phenyl-*N*-triflylimino-λ^3^-iodane, which has only recently been introduced in organic synthesis, in the presence of N-halosuccinimides as oxidants, have been investigated. Depending on the substrate, solvent, and oxidant, a variety of linear and heterocyclic products of triflamidation were obtained. Thus, in methylene chloride, the products of halotriflamidation were obtained with styrene and vinylcyclohexane, and the regioselectivity of the reaction was determined. In acetonitrile, the solvent competes with weak triflamide anion in addition to the C=C double bond and leads to the formation of amidines. The reaction with stilbene was shown to be the first example of the synthesis of *N*-triflylaziridines using *N*-triflylimino-λ^3^-iodane. Remarkably, in the reaction with NIS, *N*-triflylaziridine is formed in equally good yield in methylene chloride and in acetonitrile, while with NBS it is formed in much lower yield and only in methylene chloride; in acetonitrile, *N*-triflylimidazoline was obtained in a good yield as a result of bromotriflamidation followed by dehydrobromination with cyclization. A tentative mechanism starting with the addition of electrophilic halogen to alkene is proposed that accounts for the formation of all observed products.

## Data Availability

The data presented in this study are available on request from the corresponding author.

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
