# Peer review of "The Reactions of Alkenes with Phenyl-*N*-triflylimino-λ^3^-iodane: Solvent and Oxidant Impact"

_ijms, 2023, doi:10.3390/ijms242115947_

Round 1
Reviewer 1 Report
Comments and Suggestions for Authors
The authors presented important work in reactions of alkenes with phenyl-N-triflylimino-λ3-iodane 98 PhI=NTf (1) in different conditions.
The manuscript, after correction of some remarks, may be accepted for publication.
1. In the introduction, the authors give an overview of what is currently known in the field of study. This is important, but from the reviewer's point of view, the introduction is too long. For example, the last paragraph before the scheme is not directly related to the discussed reaction. It would have been better if instead of 4 schemes, there had been one scheme summarising the most important and relevant areas. Does it make sense to give 50 references in the introduction?
2. It is not always clear which substances are new and which have been described before, it makes sense to put this in the conclusions.
3. The scheme with the supposed mechanism does not look good. Perhaps the authors should have depicted the products instead of writing the numbers of compounds for the convenience of the readers.
4. The most interesting compound is 10. Have the authors attempted to involve this substance in new reactions?
5. Substances 13 and 17 were not isolated in pure form by the authors, but were recorded as a mixture. How correct is it to depict in the article the structure of compounds that could not be isolated in pure form? Perhaps the structures of these compounds should have been given in square brackets.
6. The caption to Scheme 11 is not correct. Thus, the same reaction without acetonitrile is shown in Scheme 10.
7. The authors need to describe the scheme 13 with the supposed mechanism more carefully and in detail. So, for example, how exactly from the intermediate cation shown below the formation of compound 20 proceeds?
Author Response
Thanks to Reviewer 1 for questions, comments and suggestions to improve our work.
- In the introduction, the authors give an overview of what is currently known in the field of study. This is important, but from the reviewer's point of view, the introduction is too long. For example, the last paragraph before the scheme is not directly related to the discussed reaction. It would have been better if instead of 4 schemes, there had been one scheme summarising the most important and relevant areas. Does it make sense to give 50 references in the introduction?
The text is revised, the first three Schemes are merged in one Scheme, the forth Scheme is omitted. The number of references is substantially reduced.
- It is not always clear which substances are new and which have been described before, it makes sense to put this in the conclusions.
All new products are fully characterized, while for the known compounds the corresponding references are given. We believe that it is enough for the inquiring reader.
- The scheme with the supposed mechanism does not look good. Perhaps the authors should have depicted the products instead of writing the numbers of compounds for the convenience of the readers.
Done. Thanks to the reviewer for the improvement.
- The most interesting compound is 10. Have the authors attempted to involve this substance in new reactions?
The Reviewer is right, indeed, this is the most interesting product but its reactivity and possible transformations will be the subject of our further investigations. Here, we show its formation as a unique precedent.
- Substances 13 and 17 were not isolated in pure form by the authors, but were recorded as a mixture. How correct is it to depict in the article the structure of compounds that could not be isolated in pure form? Perhaps the structures of these compounds should have been given in square brackets.
The products of halotriflamidation 13 and 14, as well as 16 and 17, indeed, are formed as inseparable mixtures of the corresponding regioisomers distinguished only by NMR. However, it does not influence the elemental analysis, so, the composition is unequivocally proved and, in our opinion, it doesn't make any sense and would be wrong to enclose them in square brackets.
- The caption to Scheme 11 is not correct. Thus, the same reaction without acetonitrile is shown in Scheme 10.
The reaction in Scheme 10 is not the same as the switch from CH2Cl2 to MeCN gives rise to new products. But we are thankful for this comment as it allows us to unify Schemes and to correct “cyclo-C6H13” to “cyclo-C6H11” (unnoticed by the Reviewer, sorry for the misprint).
- The authors need to describe the scheme 13 with the supposed mechanism more carefully and in detail. So, for example, how exactly from the intermediate cation shown below the formation of compound 20 proceeds?
DONE. Thanks to the Reviewer for suggestion to improve the article.

Reviewer 2 Report
Comments and Suggestions for Authors
Numbering of compounds is out of sequence and confusing. i.e. pg4, line 127-133 compound 7 in 54% yield ( I cannot find the corresponding figure).
Scheme 13 - I would replace the numbering of the structures with actual general/generic structures as otherwise it is hard to follow what is going on in the proposed mechanisms.
I'd suggest adding more context around the mechanistic pathways based on solvent polarity and potential solvent reactivity.
The method section should include a protocol for the preparation of the key reagent: phenyl-N-trifylimino-λ3 -iodane so that readers can reproduce the work fully if interested.
Comments on the Quality of English Language
I'd suggest dropping the "The Diversity of Products" as well as "Solvent and Oxidant Impact" from the title to the simpler title of "The reaction of alkenes with Phenyl-N-trifylimino-λ3-iodane.
Please find an alternative adjective other than "special" and/or re-write the 1st sentence in the intro.
The introduction is rather lengthy and could be simplified to be more concise to focus more around Phenyl-N-trifylimino-λ3-iodane. The discussion on aziridination (i.e. Scheme 1 and Scheme 3 and accompanying text) could be omitted or drastically reduced. Likewise, Scheme 4 and associated text could easily be omited.
Author Response
Thanks to Reviewer 2 for comments and suggestions to improve our work.
Numbering of compounds is out of sequence and confusing. i.e. pg4, line 127-133 compound 7 in 54% yield (I cannot find the corresponding figure).
Not all compounds must be drawn in Schemes. Product 7 is a known product of triflamidation, its formula TfNHCH(Ph)CH2NHTf 7 is given, and we believe there is no need to say more.
Scheme 13 - I would replace the numbering of the structures with actual general/generic structures as otherwise it is hard to follow what is going on in the proposed mechanisms.
DONE
I'd suggest adding more context around the mechanistic pathways based on solvent polarity and potential solvent reactivity.
DONE
The method section should include a protocol for the preparation of the key reagent: phenyl-N-trifylimino-λ3-iodane so that readers can reproduce the work fully if interested.
DONE
I'd suggest dropping the "The Diversity of Products" as well as "Solvent and Oxidant Impact" from the title to the simpler title of "The reaction of alkenes with Phenyl-N-trifylimino-λ3-iodane.
We thank the Reviewer for his suggestion. ‘The diversity of products’ is now deleted but we would like to keep “Solvent and Oxidant Impact” since it is essential and deserves mentioning in the title.
Please find an alternative adjective other than "special" and/or re-write the 1st sentence in the intro.
Done.
The introduction is rather lengthy and could be simplified to be more concise to focus more around Phenyl-N-trifylimino-λ3-iodane. The discussion on aziridination (i.e. Scheme 1 and Scheme 3 and accompanying text) could be omitted or drastically reduced. Likewise, Scheme 4 and associated text could easily be omitted.
The Introduction is now reduced. Schemes 1-3 are combined, Scheme 4 deleted.

Reviewer 3 Report
Comments and Suggestions for Authors
The principal author of this manuscript (Ganin) have reported two recent studies related to the topic of this manuscript, that is, this research group has had experience in the field.
The paper is well-organized and gives a detailed introduction of related previous works (about 50 references).
Here they describe the synthesis of a range of products with some of them were isolated in very low yields. Nevertheless, they describe the full possibility of the transformations of compound 1.
1. Remarks
p.6, Scheme 13: “A tentative mechanism for the formation…”
I do not think that it is a mechanism; this scheme simple shows the transformation possibilities discussed in the manuscript.
2. Minor remarks
i) correct forms are 61%, 75–89% etc.; similarly 0 oC (Scheme 1)
N-triflylimino - sometimes with N italic, in other cases, it is not
Schemes 5,10,11 etc.: correct the position of the vinyl group (there should be 120 degree between the sigma and pi bonds)
in a similar manner, the structure of 17, 18 etc. is quite odd?
ii) English usage
line 79: …the middle the of 70s
Author Response
Reviewer 3
Thanks to Reviewer 3 for remarks and suggestions to improve our work.
- Remarks
p.6, Scheme 13: “A tentative mechanism for the formation…”
I do not think that it is a mechanism; this scheme simply shows the transformation possibilities discussed in the manuscript.
Corrected. We added some explanations.
- Minor remarks
- i) correct forms are 61%, 75–89% etc.; similarly 0 oC (Scheme 1)
Corrected.
N-triflylimino - sometimes with N italic, in other cases, it is not
Corrected.
Schemes 5,10,11 etc.: correct the position of the vinyl group (there should be 120 degree between the sigma and pi bonds)
Corrected.
in a similar manner, the structure of 17, 18 etc. is quite odd?
We don’t understand what’s “odd” in these structures.
- ii) line 79: …the middle the of 70s
Corrected.

Round 2
Reviewer 1 Report
Comments and Suggestions for Authors
after the authors have made the necessary changes, the manuscript can be accepted for publication.